# Action Recognition Using Close-Up of Maximum Activation and ETRI-Activity3D LivingLab Dataset

**DOI:** 10.3390/s21206774

**Published:** 2021-10-12

**Authors:** Doyoung Kim, Inwoong Lee, Dohyung Kim, Sanghoon Lee

**Affiliations:** 1Department of Electrical and Electronic Engineering, Yonsei University, Seoul 03722, Korea; tnyffx@yonsei.ac.kr (D.K.); mayddb100@yonsei.ac.kr (I.L.); 2Intelligent Robotics Research Division, Electronics and Telecommunications Research Institute, Daejeon 34129, Korea; dhkim008@etri.re.kr; 3Department of Radiology, College of Medicine, Yonsei University, Seoul 03722, Korea

**Keywords:** action recognition, dataset shift, self-attention map

## Abstract

The development of action recognition models has shown great performance on various video datasets. Nevertheless, because there is no rich data on target actions in existing datasets, it is insufficient to perform action recognition applications required by industries. To satisfy this requirement, datasets composed of target actions with high availability have been created, but it is difficult to capture various characteristics in actual environments because video data are generated in a specific environment. In this paper, we introduce a new ETRI-Activity3D-LivingLab dataset, which provides action sequences in actual environments and helps to handle a network generalization issue due to the dataset shift. When the action recognition model is trained on the ETRI-Activity3D and KIST SynADL datasets and evaluated on the ETRI-Activity3D-LivingLab dataset, the performance can be severely degraded because the datasets were captured in different environments domains. To reduce this dataset shift between training and testing datasets, we propose a close-up of maximum activation, which magnifies the most activated part of a video input in detail. In addition, we present various experimental results and analysis that show the dataset shift and demonstrate the effectiveness of the proposed method.

## 1. Introduction

There have been significant improvements in action recognition researches [1,2,3,4,5,6,7,8] with various industrial applications including surveillance systems, human–computer interaction, virtual reality, sports video analysis and home-care robots [9,10,11,12,13,14,15,16,17]. The datasets such as Kinetics [1], UCF [10] and HMDB [11] are generally used public datasets in action recognition, which were made by clipping and collecting existing videos. These datasets have a great variety of videos in terms of subject, background, lighting condition and view point. The diverse characteristics could enable classifiers to learn powerful generalized representation to avoid overfitting. Nevertheless, actions involved in the datasets sometimes differ from those that we need to use in a specific application. For example, although actions such as *putting on cosmetics*, *spreading bedding/folding bedding*, and *doing neck roll exercise* are very ordinary in real life, these are not included in the datasets consisting of collected videos such as Kinetics, UCF and HMDB. Owing to this problem, despite the remarkable development of the action recognition model, it is difficult to develop the model which recognizes specific target actions in the actual industry. This lack of the target actions forces users to acquire data directly.

Due to the need for target action data, datasets containing daily activities such as NTU RGB+D [18,19], ETRI-Activity3D [20], and KIST SynADL [21] have been actively released. They were produced on purpose by directly capturing actions for actual use instead of collecting existing videos. However, the action data capture requires a fixed environment set-up for efficiency, and the fixed environment limits diversity of action videos in terms of camera angles, backgrounds, and lighting conditions. When the action recognition models are trained on these videos, the limitations lead the models to overfitting and poor generalized representation, which could not handle various characteristics of other environments such as interior design, structure, and furniture arrangement.

In the field of machine learning research, generalization issues arising from gaps between training and testing data have been studied. The issues are also very important in action recognition, but the existing datasets cannot provide sufficient data to deal with target actions with various characteristics. However, since the datasets have limited attributes in the target actions, it is difficult to test on target action videos with different attributes. To handle this problem, we introduce a new dataset of ETRI-Activity3D-LivingLab captured for testing on actual environments. The new dataset has the same class configuration as ETRI-Activity3D [20] and KIST SynADL [21], but has been produced in diverse environments different from the other datasets. Through action recognition models trained on ETRI-Activity3D [20] and KIST SynADL [21] and tested on ETRI-Activity3D-LivingLab, the performance degradation problem due to environmental changes can be observed.

In this paper, we introduce a new ETRI-Activity3D-LivingLab dataset, which is captured in an actual environment with tremendous diversity and provides useful data to users and researchers. The new dataset is available online at https://ai4robot.github.io/etri-activity3d-livinglab-en/. As shown in Figure 1, the ETRI-Activity3D-LivingLab dataset has different characteristics in terms of lighting condition, pose, background, and action scale while the ETRI-Activity3D dataset shows monotonous characteristics. These different characteristics cause dataset shift, which occurs when models encounter unseen dataset, and predicting action labels of the ETRI-Activity3D-LivingLab videos with action recognition models trained on other datasets is a very challenging task. We expect the ETRI-Activity3D-LivingLab dataset helps to study reducing the dataset shift due to different attributes of the same actions.

To reduce the dataset shift, we train the action recognition model to focus on actions by learning diverse scales of the actions. One of the main obstacles to training the ETRI-Activity3D dataset is monotonous attributes such as action scales. The scale of a foreground containing a human action in the ETRI-Activity3D videos is mostly constant for each action due to the fixed camera set-up, but the action scale appears different in testing environments. To handle the diverse action scales, action videos need to be cropped with various scales, but action information can be excluded in a video input when cropping without a guide to the position of an action. Meaningless video inputs without action information severely degrades the performance of classifiers. To prevent the classifiers from sampling meaningless parts, the close-up of maximum activation searches the action region based on self-attention map, and magnifies the action region from the video input. Through this process, the classifiers can focus more on the action in various backgrounds, and this leads to better performance for action recognition.

Finally, we present various experimental results and analysis that show the dataset shift on the ETRI-Activity3D-LivingLab dataset and demonstrate the effectiveness of the proposed method. We provide not only ablation studies according to action scales but also comparison with other methods. In particular, dataset shift analysis shows one of the reasons that the dataset shift occurs between datasets in terms of an action box detected from a video. Furthermore, we propose MaxOverlap to measure the difference of an action box between datasets and show that the improvement of MaxOverlap values by the proposed method leads to action recognition performance enhancement.

## 2. Related Works

**Action recognition datasets.** The action recognition datasets have played a very important role in the development of 3D classification tasks. The RGB-based action recognition problem has been studied for a very long time, and the datasets contain diverse situations and actions. Kuehne et al. [11] extracted clips from a variety of sources ranging from digitized movies to YouTube. Soomro et al. [10] collected videos of 101 classes that were provided to overcome the existing unrealistic environments and a small number of action classes. Researchers have used these two datasets to improve deep learning techniques for action recognition before Kinetics [1]. Carreira et al. [1] released a truly large-scale video dataset, and inflated the existing image classification network to a 3D classification network, and by learning the tremendous representation power of the dataset in advance, the action classifier showed better performance on other datasets. Although the development of deep learning techniques has been achieved, appropriate action data for actual target applications are still necessary to apply these techniques.

Due to the lack of action categories and three-dimensional information in RGB videos, datasets captured by various sensors to provide multimodal data were released. In particular, it is very easy to acquire data by using devices such as a Kinect equipped with multiple sensors [22,23]. From Kinect, Shahroudy et al. [18] and Liu et al. [19] could capture multimodal data such as RGB, depth, infrared images, and skeletons in the NTU RGB+D dataset. Despite its controlled environment, this large-scale dataset helps to improve deep learning techniques [2,24,25,26,27,28,29,30,31,32,33] with various approaches. The ETRI-Activity3D dataset [20] is also the multimodal dataset, which provides the actions of the eldery, and KIST SynADL [21] generated synthetic data with motion capture data. It was possible to recognize target actions by the release of the datasets, but there was no guarantee that it would recognize the target actions in different environments with high accuracy. In this respect, since the ETRI-Acitivity3D-LivingLab dataset contains action sequences in diverse environments with the same action class configuration with ETRI-Activity3D and KIST SynADL, the new dataset can help researchers to evaluate the generalization performance of their methods.

**Action recognition networks.** Researchers have proposed methods for applying image classification networks [34,35,36,37,38,39,40] to temporal video data. Inflated versions [1,3] of classification networks have played a role as the great baseline for action recognition with the accuracy of 75.7% and 75.4% on Kinetics. In particular, Tran et al. [3] inflated Deep Residual Network [37,38], and ResNet3D has been widely used as a core structure used in action recognition. Wang et al. [4] proposed the non-local block which comprehended non-local dependencies among spatiotemporal features and achieved the accuracy of 77.7% on Kinetics. Feichtenhofer et al. [6] proposed the SlowFast network, which fused spatial semantics at a low frame rate and temporal motion at a high frame rate and the action recognition accuracy was improved to 79.8%. Expanded architectures [8] for efficient video recognition predicted actions with reduced computational costs and fewer parameters. Only with 20% flops of the SlowFast network, X3D [8] achieved the accuracy of 79.1% on Kinetics.

**Self-attention map.** Zhou et al. [41] proposed class activation mapping, which predicts an object region without a localization label and showed the localization accuracy of 46.29% on ImageNet-1k [34]. However, the class activation map represents only a discriminative part of an image, not the overall shape of an object in the image. To overcome this limitation, Choe et al. [42] proposed an attention-based dropout layer, which randomly dropped information of the discriminative part in the image to activate an entire object region by using its self-attention map. Based on this insight, we adopt the self-attention map of an action video, which is examined in detail by cropping activated parts only. The attention-based dropout technique achieved the localization accuracy of 48.71% on ImageNet-1k.

### Contributions

We arrange the main contributions as follows:We introduce a new ETRI-Activity3D-LivingLab dataset with tremendous diversity, which does not exist in the ETRI-Activity3D [20] and KIST SynSDL [21] datasets. We reveal the dataset shift problem between datasets and propose a new training and testing protocol to target it.We utilize a self-attention map [42] to localize the activated part in a video, and the close-up of the maximum activation process magnifies a meaningful part for action recognition precisely. The proposed method mitigates the influence of the background and allows the action model to see the foreground in detail.We present various experimental results and analyses on ETRI-Activity3D [20], KIST SynADL [21], and ETRI-Activity3D-LivingLab. The experimental results demonstrate that the close-up of the maximum activation process effectively reduces the dataset shift problem by using various methods.

## 3. System Model

In this section, we describe our system model consisting of two main steps, maximum activation search and a close-up of the maximum activation. A detailed description of the proposed model is provided, and we specify only the spatial size of a video for convenience because the temporal length of the video input is fixed in the system model.

### 3.1. Overview of System Model

As shown in Figure 2, the proposed method is composed of the first and second flows. The first flow is the process of obtaining the feature map of the initial video input $I_{0}\left(x,y,t\right)$, and the second flow is the action recognition process to search and magnify the most activated region of the initial video input. Specifically, in the first flow indicated by the solid lines, the original video is resized to the initial size $W_{0}\times H_{0}$ and randomly cropped (*RandCrop*) to always contain its human action, and the backbone network is trained with the initial video input for action classification. In the second flow indicated by the dotted lines, the self-attention map A(x,y) is obtained from the feature map of the initial video input in the first flow for the maximum activation search. In the maximum activation search, the most activated region for action classification is searched with the kernel of a size K×K, and the kernel size and position $\left(x_{ma},y_{ma}\right)$ of the region are delivered to the next step. In the close-up of maximum activation, the original video is resized to $W_{K}\times H_{K}$ and the cropping position $\left(x_{K},y_{K}\right)$ is calculated from *K* and $\left(x_{ma},y_{ma}\right)$. The resized video is cropped in the position $\left(x_{K},y_{K}\right)$ (*ActCrop*), and the backbone network is trained with this magnified video input $I_{K}\left(x,y,t\right)$ for action classification.

### 3.2. Maximum Activation Search

As the position of an action in a video is unknown, the backbone network needs to watch the video globally and find the position of the action. To always contain the action regardless of the cropping position, the video is resized spatially to the sufficient size $W_{0}\times H_{0}$ and randomly cropped (*RandCrop*) to the size L×L. *RandCrop* is one of data augmentation techniques [3] and enables the networks to learn different video inputs from the same video which has fixed backgrounds in the ETRI-Activity3D dataset. *RandCrop* also has an influence on magnified video inputs $I_{K}\left(x,y,t\right)$ due to the random position for cropping and can slightly improve the classification performance. The backbone network is trained with this initial video input $I_{0}\left(x,y,t\right)$ for action classification. The action region is activated in the self-attention map A(x,y) obtained by taking the average along the temporal and channel axes of the feature map generated by the backbone network in (1) Maximum Activation Search of Figure 2.

In this self-attention map, to search the maximum activation for action classification, average-pooling is used with a kernel of size K×K, where *K* has a value from 3 to 6, and the position of the maximum activation $\left(x_{ma},y_{ma}\right)$ of Figure 3a is obtained as shown in the light blue box of Figure 2. Therefore, the maximum activation search in the self-attention map is defined as
(1)$$\left(x_{ma},y_{ma}\right)=\underset{\left(x,y\right)}{\mathrm{argmax}}AvgPool_{K\times K}\left(A\left(x,y\right)\right).$$

### 3.3. Close-Up of Maximum Activation

From the maximum activation search, the kernel size and the position of the maximum activation are obtained. To magnify the initial video input $I_{0}\left(x,y,t\right)$, the original video is resized to $\left(W_{K},H_{K}\right)$ according to the kernel size K×K. As shown in Figure 3b, the magnified video input is $\frac{L}{PK}$ larger than the initial video input, where *P* is the pooling factor by the backbone network. According to this ratio $\frac{L}{PK}$, the position of the maximum activation $\left(x_{K},y_{K}\right)$ is calculated in the close-up and defined as
(2)$$\left(x_{K},y_{K}\right)=\left(\frac{x_{ma}L}{PK},\frac{y_{ma}L}{PK}\right).$$

The resized video input is also cropped to the size L×L in this position of Equation (Equation 2) (*ActCrop*). Since the cropping size is same and the resized video input is enlarged, the activated region is focused and magnified with the same cropping size. The backbone network is trained with this magnified video input $I_{K}\left(x,y,t\right)$ and the initial video input $I_{0}\left(x,y,t\right)$ together for action classification.

### 3.4. Training and Testing

**Training phase.** The proposed close-up of maximum activation provides a magnified video input which consists of meaningful motion for action recognition only in training. By learning the initial video inputs and the magnified video inputs together, the classifier is able to handle various action scales. To help understanding, we describe the detailed training process of the proposed method in Algorithm 1.

**Testing phase.** Since various action scales are learned in training, in testing, a video input is cropped in the center position and predicted by the classifier without adjusting an action scale. Nevertheless, we examine how an accuracy changes depending on the actions scales in experimental results.
**Algorithm 1:** Training a video input by using the close-up of maximum activation.
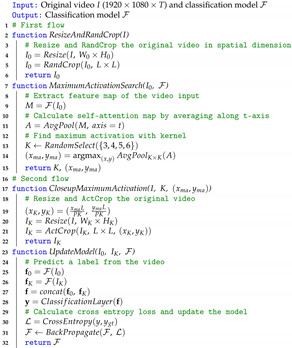


## 4. Datasets

In this section, we introduce three datasets, ETRI-Activity3D [20], KIST SynADL [21], and ETRI-Activity3D-LivingLab. These datasets share the same action classes, but have different subjects and environments. These differences make it difficult to transfer the knowledge learned from one dataset to another. We describe the detailed explanations of the three datasets as follows.

**ETRI-Activity3D** [20]. This dataset was performed by actors/actresses and captured to recognize 55 daily activities in a robot view. The total number of sequences is 112,620, and the number of subjects is 100, which include half and half each for senior citizens and young adults. This dataset provides an action sequence in eight views of various heights and angles using Kinects, but the fixed positions of Kinects result in limited action scales and backgrounds. These limitations lead the classifier to overfitting, resulting in poor action recognition performance in other datasets.

**KIST SynADL** [21]. The daily activities in this dataset were recorded by motion capture (MoCap), and randomized textures, viewpoints, and lighting conditions were added on top of a static real-world background image to generate data. Synthetic video can be generated from various camera viewpoints by utilizing the advantages of a game engine. The total number of sequences is 462,200, and they generated data with 15 characters and 4 backgrounds. However, despite the large amount of video data, the texture of computer graphics is one of the factors that degrade action predictions in actual environments. This dataset shift makes it difficult for action classifiers to correctly predict the action of a real video.

**ETRI-Activity3D-LivingLab.** Although two aforementioned datasets were released, they are still limited in terms of lighting condition, pose, background, and action scale. Therefore, it is questionable whether it is possible to recognize human action captured in a completely different environment from ETRI-Activity3D and KIST SynADL. To solve this question, we introduce the ETRI-Activity3D-LivingLab dataset, which captures daily activities in real elderly homes. A total of 30 elderly people participated, and the total number of video sequences is 6,589. For each action sequence, the videos are captured in 2 to 12 views. This dataset provides multimodal data such as RGB, depth, and skeletons using Kinect v2. Figure 1 shows various characteristics such as lighting condition, pose, background, and action scale due to the different environments of each house for the action like *Eating food with a fork*. The ETRI-Activity3D-LivingLab dataset is very useful for evaluating the generalization performance of action recognition models because it is composed of an actual environment completely independent of the existing datasets.

## 5. Implementation Details

The initial size of the video input $T\times W_{0}\times H_{0}$ is 16×171×128, and the network input size is equal to the cropping size, L=112, which is the same value used in [3]. Random sampling with a stride of 4 is used in both training and testing in the temporal dimension, and random and center cropping are used in training and testing in the spatial dimension, respectively. The network used for action recognition is ResNet3D-18 [3], which is modified in temporal pooling and downsampling of skip connection layers. The videos are not pooled in temporal dimension to keep the temporal information, and max-pooling layers with stride 2 and convolutional layers with stride 1 are used instead of convolutional layers with stride 2 in the spatial dimension. This modification yields higher accuracies by 2–3%. When training with ResNet3d-34 and 50, there is no improvement in action recognition accuracy. The scale factor of pooling *P* is 16, and the spatial size of the feature map $\frac{L}{P}\times \frac{L}{P}$ is 7×7. The SGD optimizer is used with the cosine annealing scheduler. The initial learning rate, momentum, and weight decay are 0.01, 0.9, and −5, respectively. The network is trained on seven GPUs and each GPU has eight clips in a mini-batch (so in total with a mini-batch size of 56 clips).

Baseline model trains only one video size, 171×128 without the close-up of maximum activation. For the close-up of maximum activation, the size of the average-pooling kernel in the maximum activation search is ranged from 3×3 to 6×6. Corresponding video sizes in the close-up of maximum activation are 200×149, 239×179, 299×224, and 388×299, respectively. Including the scale in maximum activation search, the network can learn videos of five scales simultaneously. In training, the initial size and the randomly selected scale are used together, and the mid-level scale is used for testing. For example, in the case of 5 scales, the videos of size 239×179 is used for the final predictions.

## 6. Experimental Results

In this section, we evaluate the action recognition performance of the proposed method and other methods on the ETRI-Activity3D-LivingLab dataset with various training data combinations. Through these experiments, we investigate practical problems of action recognition in terms of dataset shift.

### 6.1. Intra-Dataset (ETRI-Activity3D)

We evaluate the proposed method according to protocols, cross-subject (CS) and cross-view (CV), which are provided in [20,21], respectively. These are intra-dataset protocols, where training and testing data are from the same dataset. In the CS protocol, 67 subjects are used for the training set {1,2,4,5,…,97,98,100} and 33 subjects are used for the testing set {3,6,9,…,93,96,99}. In the CV protocol, 6 camera views are used for the training set {3,4,5,6,7,8} and 2 camera views are used for the testing set {1,2}.

Table 1 shows the action recognition results of the existing methods [2,20,43] by using RGB or skeleton inputs and the proposed method according to the CS and CV protocols. Since the training and testing data are from the same dataset, all methods achieve more than 80% accuracy. Specifically, the baseline and proposed method show much higher accuracy of 95% by using only RGB videos than the other methods. The classifiers seem to work successfully for action recognition, but it is unclear that they also work well in other environments which differ from those of the ETRI-Activity3D dataset. In order to resolve this question, the ETRI-Activity3D-LivingLab dataset is introduced for testing and consists of videos with unfamiliar characteristics to the classifiers.

### 6.2. Inter-Dataset (ETRI-Activity3D and ETRI-Activity3D-LivingLab)

To measure the performance of action recognition models in real-life situations, we need to test the models in various environments as possible. The ETRI-Activity3D-LivingLab dataset provides real-life videos to test the models practically outside lab environments. In this sense, classifiers are trained on the ETRI-Activity3D videos and tested on the ETRI-Activity3D-LivingLab videos, and this result gives performance degradation due to their different characteristics. This is an inter-dataset protocol, where training and testing data are from different datasets.

Comparing the baseline results of Table 1 and Table 2, the action recognition accuracy is greatly degraded from 95.7% to 50.4% according to testing data. This result shows that the classifier trained on the videos in the lab environments cannot handle unfamiliar characteristics of the testing videos, and this gap occurs when action classifiers recognize target actions that have not been encountered. The performance degradation can be improved by learning different types of video data or reducing dataset shift. As it is practically difficult to collect different action videos that can occur in real life, the proposed method focuses on reducing dataset shift in terms of action scales and improves accuracy by up to 10%.

To show the effectiveness of the proposed method, we present ablation studies in Table 2, which shows the results of Baseline, *RandCrop*, and *ActCrop*. We train ResNet3D-18 network by using *RandCrop* and *ActCrop* with multiple scales. *RandCrop* means that a video of the size $W_{K}\times H_{K}$ is cropped to a magnified video input in a random position instead of two processes in the second flow of Figure 2, and *ActCrop* means the close-up process of the proposed method as shown in Figure 2. The results of *RandCrop* and *ActCrop* show that searching the position where the action occurs is very important when training multiple scales of videos in the ETRI-Activity3D datasets. *ActCrop* has a higher accuracy than *RandCrop* by up to 9.6%. Since *RandCrop* may sample the region where there is no action on the larger scale, it is tough to improve action recognition performance.

Figure 4 shows the different results between *RandCrop* and *ActCrop*. Since RandCrop literally crops videos in a random position, meaningful parts which provide a clue to action recognition may be eliminated while *ActCrop* reflects the semantics of the video and helps to handle significant motion precisely. For example, to recognize *01_eating food with a fork*, a classifier needs to take a closer look at hand motion in the video. While *RandCrop* sometimes crops in backgrounds, *ActCrop* based on the self-attention map crops exactly in the position where hand motion occurs. Incorrect cropping degrades action recognition performance by providing video inputs without motion information. Therefore, *ActCrop* plays an important role in action recognition which needs to search action in diverse scales without localization labels.

The proposed method does not improve all tested activities. After applying the proposed method with 5-scale, the classification accuracy of 10 activities decreases by 4.5% on average while the accuracy of the other activities increases by 12.9% compared to the baseline. Since the performance improvement is overwhelming rather than the performance degradation, the proposed method achieves better action recognition accuracy than the baseline. The performance degradation of 10 activities seems to be caused by underfitting. While the baseline reduces sufficiently cross entropy loss under 0.01, the proposed method could not reduce the loss under 0.54. Nevertheless, the proposed method gives better feature representation by learning the close-up of maximum activation. A more complicated model with the proposed method could not be tried due to the limitation of the hardware.

There are other factors to degrade classification performance: interaction with other objects, partially occluded objects, and low illumination. Forty activities interact with an object and the other activities are bare hands without an object. The activities interacting with an object (54.5%) achieve lower accuracy than the other activities (74.6%) because the interacting activities with an object mostly has slight motion compared to the other activities. The performance improvement by the proposed method is larger in the interacting activities with an object (+11.5%) than in the other activities (+4.3%).

The partial occlusion of objects usually occurs in cooking activities such as *08_using a gas stove* and *09_cutting vegetable on the cutting board* because cameras capture the elderly’s back in these activities. Two activities have very low accuracy, 11.7% and 5.0%, in the baseline because of the occlusion. In this case, the classifier needs to predict the activity label from small motion or background information. After applying the proposed method, the accuracy is improved to 32.5% and 25.0%, respectively.

The low illumination makes it difficult to recognize activities correctly. ‘P201’, one of the subjects in the ETRI-Activity3D-LivingLab videos, were captured in environments under relatively low illumination compared to the others. In the baseline, ‘P201’ and the others achieve the accuracy of 43.6% and 52.3%. This result shows that the low illumination degrades the action recognition accuracy. In the proposed method, the accuracy is improved to 54.6% and 61.3%, respectively.

### 6.3. Dataset Shift Analysis: ETRI-Activity3D and ETRI-Activity3D-LivingLab

As mentioned in the previous Section 6.2, we try to infer the influence of the dataset shift with the results of Table 1 and Table 2. The inter-dataset results in Table 2 are approximately 35% lower in accuracy than the intra-dataset results in Table 1. This difference shows a severe shift between the ETRI-Activity3D and ETRI-Activity3D-LivingLabdatasets. Since the homogeneous videos from the ETRI-Activity3D dataset are used and share similar data distributions in both training and testing, the classifiers predict their action labels easily due to the same characteristics of videos such as camera angles, action scales, backgrounds, and so on. On the other hand, training and testing videos used in Table 2 are heterogeneous, and this causes significant degradation in accuracy due to the different characteristics between training and testing videos.

The dataset shift is observed in detail as shown in Figure 5, which presents action class accuracies. The blue and orange bars (intra-dataset and homogeneous) indicate testing accuracies on ETRI-Activity3D according to CS and CV protocols, respectively, and the red bars (inter-dataset and heterogeneous) indicate testing accuracies on ETRI-Activity3D-LivingLab. All accuracies on the ETRI-Activity3D-LivingLab videos are much lower than those on the ETRI-Activity3D videos. In comparison with the intra-dataset results, *44_Taking a bow* and *48_Fighting each other* in the inter-dataset results have only 5–8% degradation in accuracy, while *01_Eating food with fork* and *09_Cutting vegetable on the cutting board* in the inter-dataset results have the 77–88% degradation in accuracy. Although the former actions are captured in different camera views, the poses and motions by two people are very similar in videos of both datasets, but the latter actions have different action scales and motions in their videos according to their camera views. The proposed method improves their degraded performance of *01_eating food with fork* and *09_cutting vegetable on the cutting board* by 20–28% in accuracy, and this improvement shows that the proposed method has a great effect on the actions with small motions.

To help understanding, the examples of the aforementioned actions as shown in Figure 6 are presented from two datasets. Figure 6a,b where only hand motions occur show that the actions are captured smaller in the ETRI-Activity3D videos than the ETRI-Activity3D-LivingLab videos. These actions are too small to be recognized by the Baseline network and their accuracies on ETRI-Activity3D-LivingLab are less than 10%. By magnifying these actions similar to the actions of ETRI-Activity3D-LivingLab in training, the proposed method achieves performance improvements by 25–39% in accuracy. Nevertheless, the actions have only 30–40% accuracy because there are still other factors to degrade the performance besides the action scale. Although the shift due to the action scale is reduced, the datasets have different characteristics such as light condition, pose, and background as mentioned earlier. On the other hand, Figure 6c,d where two people interact with each other have similar action scales and poses. These similar characteristics led to the high accuracy of more than 90%.

In this respect, the action scale is one of the important factors in action recognition, and we analyze action scales of videos according to an action class as shown in Figure 7, which shows the distributions of action boxes detected in the ETRI-Activity3D and ETRI-Activity3D-LivingLab videos by Yolo v3 [45]. An action box is determined by finding the maximum size which includes all human boxes detected in an action video. In general, the action boxes of ETRI-Activity3D-LivingLab is mostly larger than those of ETRI-Activity3D. These distributions mean that there is the obvious dataset shift between two datasets due to the different action scales, which cause the performance degradation. This degradation depends on an action class. The different scales of action boxes in Figure 7a,b result in the lowest accuracy while the similar scales of action boxes in Figure 7c,d result in the highest accuracy. In the proposed method, the action boxes of ETRI-Activity3D-LivingLab videos are mostly similar to those of magnified ETRI-Activity3D videos and this leads to the performance improvement.

In terms of the action scale, the decreasing accuracy according to the increasing video size in Table 2 can be explained. ETRI-Activity3D-LivingLab videos have larger action scales than ETRI-Activity3D videos without the close-up as shown in Figure 7, and the action scale increases along with the increase of a video size because a video input is always cropped by the fixed size of L×L. The action scale of both ETRI-Activity3D and ETRI-Activity3D-LivingLab videos are enlarged by the close-up process. After the close-up, the action scale of ETRI-Activity3D-LivingLab videos is still larger than the action scale of ETRI-Activity3D videos. This action scale difference leads to the lower accuracy for the larger video size in Table 2.

To verify the influence of the action box similarity numerically, MaxOverlap (maximum overlap) is measured between action boxes of ETRI-Activity3D and ETRI-Activity3D-LivingLab and defined as:(3)$$MaxOverlap_{i}=\underset{j}{\mathrm{argmax}}\left(\frac{min(h_{liv}^{i},h_{ea3}^{j})}{max(h_{liv}^{i},h_{ea3}^{j})}*\frac{min(w_{liv}^{i},w_{ea3}^{j})}{max(w_{liv}^{i},w_{ea3}^{j})}\right),$$
where $h_{liv}^{i}$ and $w_{liv}^{i}$ is the height and width of an action box in *i*th ETRI-Activity3D-LivingLabLivingLab video and $h_{ea3}^{j}$ and $w_{ea3}^{j}$ is the height and width of an action box in *j*th ETRI-Activity3D video. Through Equation (Equation 3), the action box in the ETRI-Activity3D video which has the maximum overlap with the action box in $i^{th}$ ETRI-Activity3D-LivingLab video is searched, and this could be a measure of how similar the action is in terms of the action scale. Table 3 shows MaxOverlap proportions for the ETRI-Activity3D-LivingLab videos. As shown in Figure 7, when using the baseline, *01_Eating food with fork* and *09_Cutting vegetable on the cutting board* with the lowest accuracy have relatively low MaxOverlap values while *44_Taking a bow* and *48_Fighting each other* with the highest accuracy have about 90% of MaxOverlap values in the interval between 0.9 and 1.0. When the proposed method is applied, all actions have high MaxOverlap values, which are concentrated on the interval between 0.9 and 1.0 with the performance improvement as mentioned in Section 6.2. From these proportions, a large MaxOverlap value can be expected to reduce the impact of dataset shift due to the action scale and enhance an action recognition accuracy.

### 6.4. Inter-Dataset (KIST SynADL, ETRI-Activity3D, and ETRI-Activity3D-LivingLab)

Since computer graphics enable to generate data with diverse characteristics, a synthetic video dataset is captured in various camera angles, appearances, lightings, and backgrounds. KIST SynADL were designed to complement limited characteristics of ETRI-Activity3D, and we expect the synthetic data to improve the performance on the ETRI-Activity3D-LivingLab videos. However, it is difficult to show good performance by learning the synthetic dataset alone because there is domain shift between KIST SynADL and ETRI-Activity3D-LivingLab. In this section, we demonstrate step by step how the KIST SynADL videos can help to improve action recognition performance on the ETRI-Activity3D-LivingLab dataset and also provide an analysis of action boxes.

As shown in Table 4, the classifiers are trained on the KIST SynADL or ETRI-Activty3D videos and tested on the ETRI-Activity3D-LivingLab videos. (1) KIST SynADL is trained only in a supervised manner. (2) KIST SynADL (S) + ETRI-Activity3D (T) are trained by Unsupervised Domain Adaptation (UDA) [46], which learns action semantics from a source domain and domain characteristics from a target domain simultaneously. (3) ETRI-Activity3D is trained only in a supervised manner and is reported in Table 2. (4) KIST SynADL + ETRI-Activity3D are trained together in a supervised manner. The testing dataset is all the same as ETRI-Activity3D-LivingLab.

Due to the dataset shift caused by the domains, (1) KIST SYnADL achieves only 32.5% and 34.2% in accuracy. To reduce the influence of the domain shift, ref. [46] is employed for UDA. The combined training, (2) KIST SynADL (S) + ETRI-Activity3D (T), improves the action recognition performance of the classifiers on the ETRI-Activity3D-LivingLab videos by about 9–10% in accuracy, and the performance improvement is still small even though the proposed method is applied. (3) ETRI-Activity3D is the same result of Table 2 and gives better accuracy than (1) KIST SynADL. Lastly, (4) KIST SynADL + ETRI-Activity3D gives the highest accuracy with improvements of approximately 9–13% over UDA. The proposed method also achieves a performance improvement of 5.6% in accuracy. These results of our dataset configuration show that the domain shift seems to be an important factor in the performance improvement.

### 6.5. Dataset Shift Analysis: KIST SynADL, ETRI-Activity3D, and ETRI-Activity3D-LivingLab

In this section, to give analysis of the action box for (1) KIST SynADL and (4) KIST SynADL + ETRI-Activity3D, we show action box distributions of *13_Wiping face with a towel*, *18_Putting on a jacket*, *48_Fighting each other*, and *52_Opening the door and walking in*, which have the most improved action recognition accuracy by training ETRI-Activity3D videos additionally.

In Figure 8, the distributions of the action boxes detected from original videos and magnified videos are presented to compare the results of the baseline and the proposed method. Even though the action boxes of KIST SynADL and ETRI-Activity3D-LivingLab videos overlap a lot, there is only an improvement of 1.7% in accuracy as shown in Table 4. In the case of *48_Fighting each other*, both classifiers give a zero accuracy although their *MaxOverlap* values increase obviously after applying the proposed method in Table 5. It seems that the domain shift between KIST SynADL and ETRI-Activity3D-LivingLab is very dominant despite applying the proposed method. The domain adaptation method [46] gives the improvement from 0% to 6% in accuracy for *48_Fighting each other*, but the severe domain shift still remains. Consequently, if the domain shift is not handled, there are limits to improving action recognition accuracy with the proposed method.

By training both KIST SynADL and ETRI-Activity3D in a supervised manner, the different action scales lead to the performance improvement as shown in Table 4. As shown in Figure 9, the action boxes (red samples) of ETRI-Activity3D provide additional action scales which the action boxes (blue sample) of KIST SynADL do not cover, and this complement leads to the great improvement in action recognition accuracy. Especially, the result of the baseline shows the improvement greatly to 86.6% in accuracy for *48_Fighting each other* by learning the KIST SynADL and ETRI-Activity3D videos together. Furthermore, the proposed method achieves 93.3% in accuracy for *48_Fighting each other*. These results show that learning data from different domains in a supervised manner helps to overcome the domain shift and maximize the effectiveness of the proposed method.

*MaxOverlap* is measured numerically for different action scales as shown in Table 5. The action boxes of KIST SynADL and ETRI-Activity3D-LivingLab are similar after applying the proposed method in terms of *MaxOverlap*. Nevertheless, the action recognition accuracy is not improved by the proposed method because it seems that there remains the domain shift between KIST SynADL and ETRI-Activity3D-LivingLab. While training both KIST SynADL and ETRI-Activity3D in supervised manner and reducing the domain shift, MaxOverlap values are improved significantly and this leads to the improvement in accuracy.

### 6.6. Application to Other Methods

To examine the effect of the proposed method, the proposed method is applied to the non-local neural network [4] and the SlowFast network [6], which are state-of-the-art action recognition models. The baseline is pre-trained without non-local blocks, and non-local blocks are trained together after the pre-training. For the SlowFast network, we use 16 frames in the fast pathway and 4 frames in the slow pathway. We train the networks on ETRI-Activity3D and evaluate the performance of the networks on the ETRI-Activity3D-LivingLab dataset.

As shown in Table 6, the applying the proposed method improves the performance of two state-of-the-art action recognition models by up to 10–12% in the accuracy. These results show that the proposed close-up of maximum activation can easily enhance the performance of action recognition models by cropping activated parts of a video.

### 6.7. Visualization of Self-Attention Map

The self-attention map in action recognition indicates activated parts where action occurs and is used to search an action region in the proposed method. In Figure 10, the self-attention map is utilized to visualize the performance improvement of the proposed method. The baseline classifier activates wider parts as shown in Figure 10a,b or wrong parts as shown in Figure 10c compared to the proposed method. On the other hand, the proposed method activates the action parts precisely and exactly compared to the baseline classifier and improves the action recognition performance. This means that the proposed method including the close-up of maximum activation makes the classifier predict exactly in terms of quantity and quality without additional network parameters and localization labels.

## 7. Conclusions

In this paper, we introduced the new ETRI-Activity3D-LivingLab dataset which can be used for generalization test and proposed the close-up of maximum activation which magnifies helpful parts for action recognition. Furthermore, through various experiments, we examined the data shift among the three datasets, ETRI-Activity3D, KIST SynADL, and ETRI-Activity3D-LivingLab. By utilizing these datasets, researchers can challenge the reduction of the dataset shift, and we expect that this study will be helpful in developing action recognition models for generalized representation. The close-up of maximum activation is a very simple method, but it can effectively improve the action recognition accuracy. While the proposed method predicts an action separately on each scale, we will research to achieve optimal performance by integrating close-up scales.

## Figures and Tables

**Figure 1 sensors-21-06774-f001:**
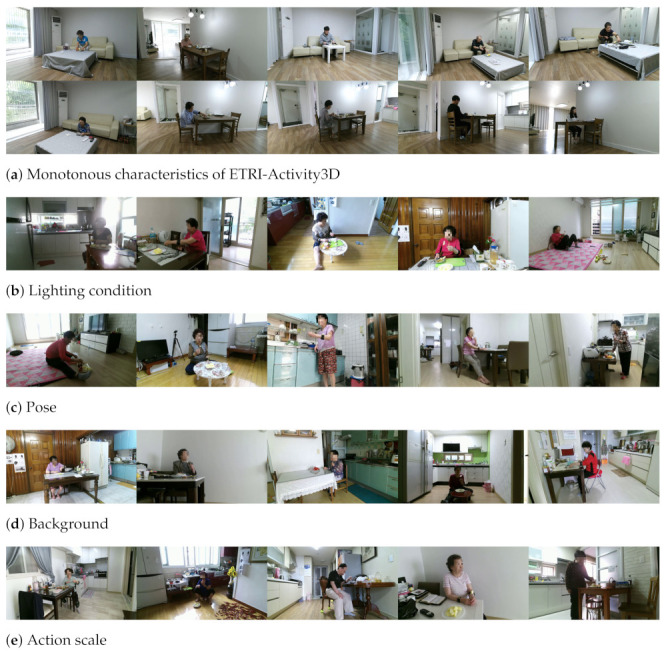
Examples of videos on *01_eating food with fork* in ETRI-Activity3D and ETRI-Activity3DLivingLab. (**a**) Shows monotonous characteristics of ETRI-Activity3D due to fixed environments while (**b**–**e**) show diverse characteristics of ETRI-Activity3D-LivingLab in terms of lighting condition, pose, background, and action scale, respectively.

**Figure 2 sensors-21-06774-f002:**
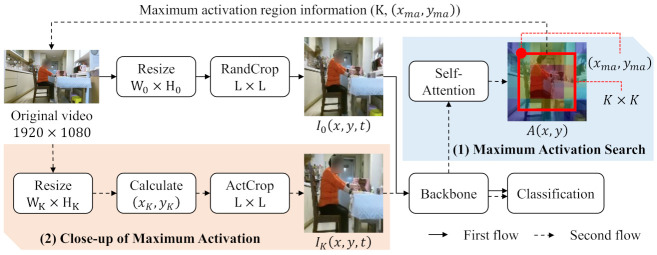
Overall system model, which consists of the maximum activation search (light blue) and the close-up of maximum activation (light orange). In the maximum activation search process, the most activated region, the red box in the self-attention map A(x,y), is represented as the action region where the action occurs. In the close-up process of maximum activation, the most activated region of the initial video input $I_{0}\left(x,y,t\right)$ is enlarged to the magnified video input $I_{K}\left(x,y,t\right)$, where *K* is the kernel size. The backbone network is trained with these two video inputs for action classification. *RandCrop* and *ActCrop* mean random crop and activation crop, respectively.

**Figure 3 sensors-21-06774-f003:**
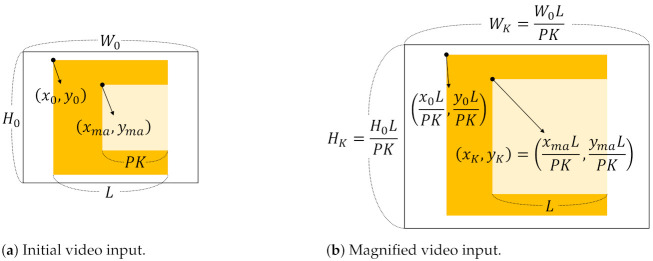
Video resizing and cropping process of the initial video input and the magnified video input. (**a**) is the initial video input for the maximum activation search, (**b**) is the magnified video input for the close-up of maximum activation. The dark yellow square and the light yellow square are the video inputs, $I_{0}\left(x,y,t\right)$ and $I_{K}\left(x,y,t\right)$, respectively. The initial video input is randomly cropped in the position (*x*_0_, *y*_0_). After the maximum activation information is obtained, the position of (*x_K_*, *y_K_*) is calculated for the close-up.

**Figure 4 sensors-21-06774-f004:**
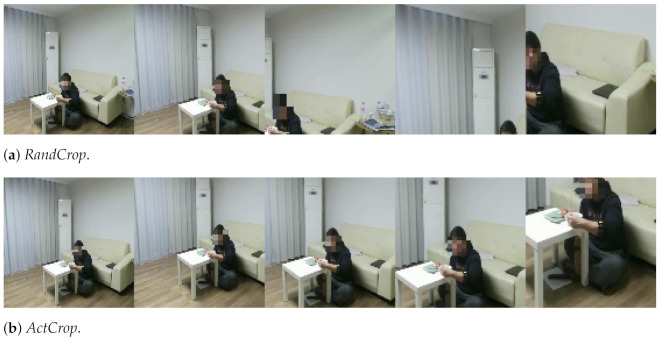
Cropping results of *RandCrop* and *ActCrop* for *01_eating food with a fork* from 1 to scale (**leftmost**) to 5-scale (**rightmost**). *RandCrop* literally crops a video input in a random position and ActCrop crops a video input in the most activated position based on its self-attention map.

**Figure 5 sensors-21-06774-f005:**
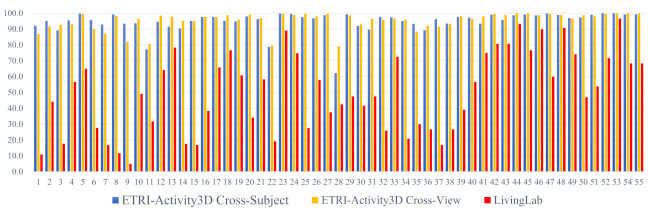
Action recognition accuracy of each action category with baseline. The results of ETRI-Activty3D CS, CV, and ETRI-Activity3D-LivingLab are presented. The x-axis and y-axis are the index of an action class and its accuracy, respectively.

**Figure 6 sensors-21-06774-f006:**
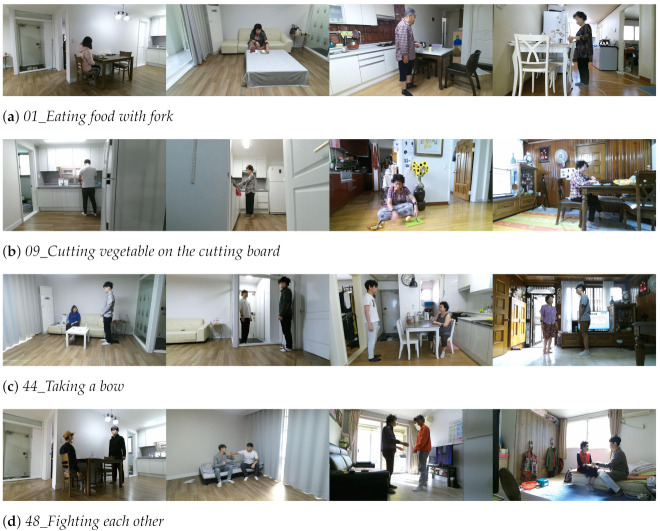
Examples of action videos that have different or similar characteristics between ETRIActivity3D (the first and second columns) and ETRI-Activity3D-LivingLab (the third and fourth columns). (**a**,**b**) show different characteristics and (**c**,**d**) show similar characteristics between two datasets.

**Figure 7 sensors-21-06774-f007:**
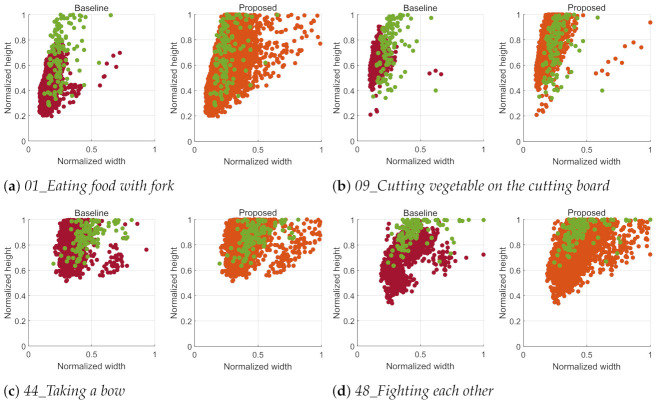
Distributions of action boxes for the same actions in Figure 6. The red and green samples are action boxes detected in ETRI-Activity3D and ETRI-Activity3D-LivingLab videos. The orange samples are action boxes of the magnified video inputs. The x-axis and y-axis are the normalized width and height of an action box, which is divided by the width and height of a video. The maximum action box size is selected to contain all human boxes detected in a video by Yolo v3 [45].

**Figure 8 sensors-21-06774-f008:**
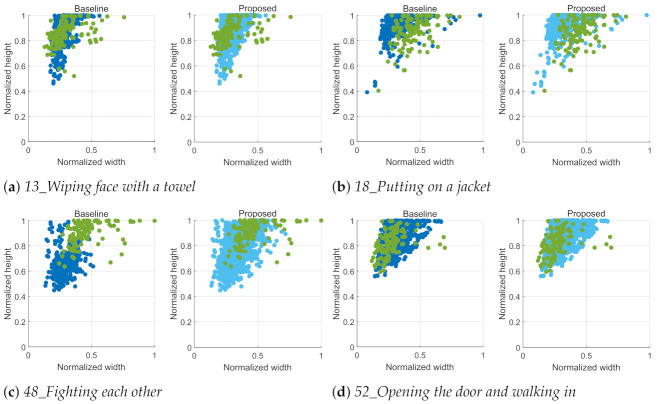
Distributions of action boxes for *13_Wiping face with a towel*, *18_Putting on a jacket*, *48_Fighting each other*, and *52_Opening the door and walking in*. The blue and green samples are action boxes detected in KIST SynADL and ETRI-Activity3D-LivingLab videos. The azure samples are action boxes of the magnified video inputs. The x-axis and y-axis are the normalized width and height of an action box, which is divided by the width and height of a video. The maximum action box size is selected to contain all human boxes detected in a video by Yolo v3 [45].

**Figure 9 sensors-21-06774-f009:**
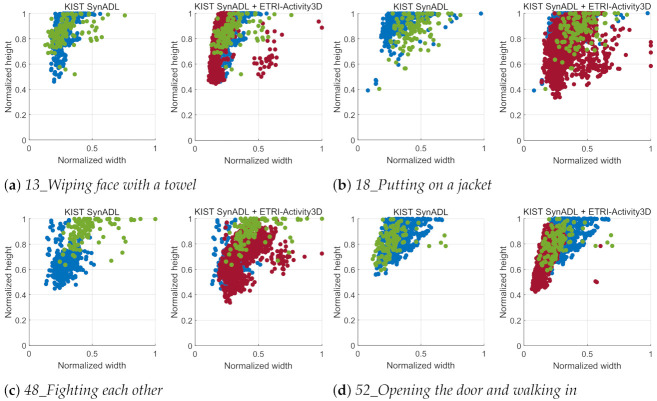
Distributions of action boxes for *13_Wiping face with a towel*, *18_Putting on a jacket*, *48_Fighting each other*, and *52_Opening the door and walking in*. The blue, red, and green samples are action boxes detected in KIST SynADL, ETRI-Activity3D and ETRI-Activity3D-LivingLab videos, respectively. The x-axis and y-axis are the normalized width and height of an action box, which is divided by the width and height of a video. The maximum box size is selected to contain all human boxes detected in a video by Yolo v3 [45].

**Figure 10 sensors-21-06774-f010:**
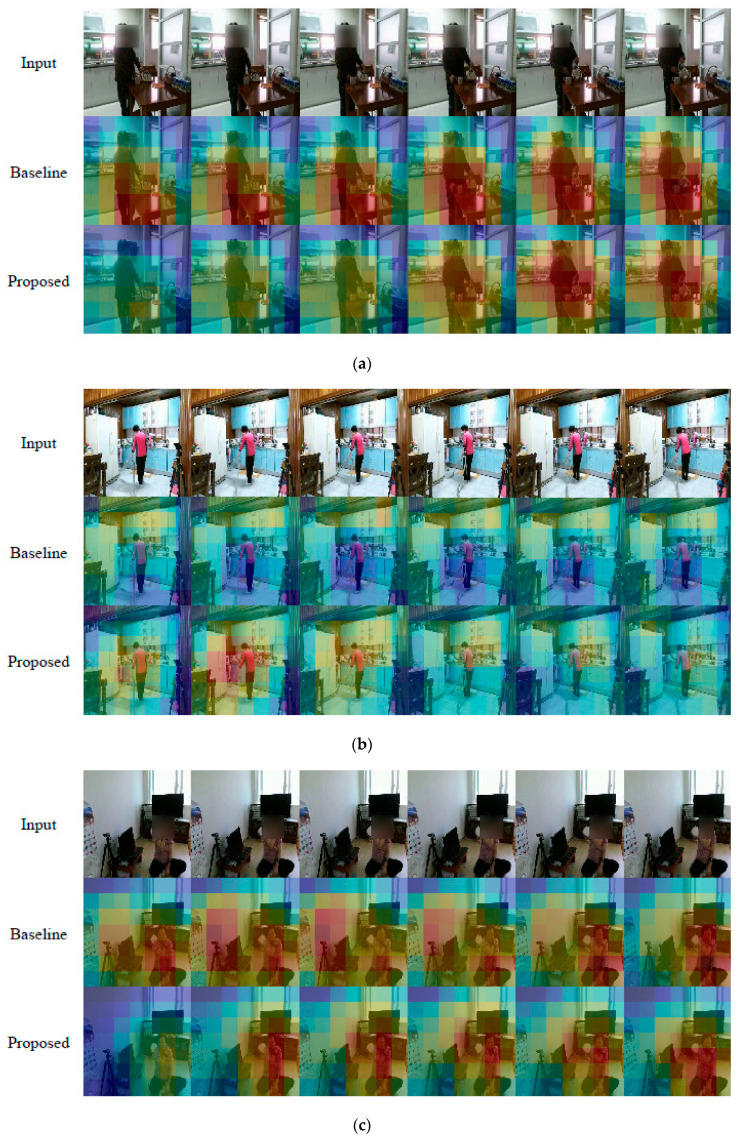
Self-attention maps on video frames activated by the baseline and the proposed method of Table 2. The red and blue color mean high and low activations, respectively. The proposed method activates correctly compared to the baseline. As a result, the baseline misclassifies (**a**), (**b**), and (**c**) as *30_looking around for something*, *45_talking to each other*, and *05_putting food in the fridge/taking food from the fridge*, respectively, while the proposed method classifies correctly.(**a**) *02_Pouring water into a cup*. The proposed method activates more precisely than the baseline; (**b**) *23_Vacuumming the floor*. The proposed method activates correctly while the baseline activates the wrong parts; (**c**) *43_Massaging a shoulder oneself*. The proposed method activates the human only while the baseline activates both the left parts and the human.

**Table 1 sensors-21-06774-t001:** Action recognition accuracies on ETRI-Activity3D according to cross-subject and cross-view protocol. The results of FSA-CNN [20] were reported on the cross-subject protocol only.

Method	Modalities	CS	CV
Glimpse [2]	RGB	80.2%	80.0%
ST-GCN [44]	Skeleton	83.4%	77.9%
VA_CNN [43]	Skeleton	82.0%	79.7%
FSA-CNN [20]	RGB	90.1%	-
	Skeleton	90.6%	-
	RGB + Skeleton	93.7%	-
Baseline	RGB	95.0%	95.3%
Proposed	RGB	95.5%	95.7%

**Table 2 sensors-21-06774-t002:** Ablation studies trained on ETRI-Activity3D and tested on ETRI-Activity3D-LivingLab. To show the effectiveness of the proposed method, *RandCrop* is used instead of *ActCrop* of (2) Close-up of Maximum Activation in Figure 2 for comparison. The accuracies of *RandCrop* and *ActCrop* from 1-scale to 5-scale are presented.

Method		171×128	200×149	239×179	299×244	399×299
Baseline		50.4%	-	-	-	-
2-scale	*RandCrop*	54.1%	50.1%	-	-	-
	*ActCrop*	54.4%	51.0%	-	-	-
3-scale	*RandCrop*	53.7%	52.5%	47.5%	-	-
	*ActCrop*	58.1%	56.6%	48.8%	-	-
4-scale	*RandCrop*	54.7%	53.4%	51.4%	44.4%	-
	*ActCrop*	58.8%	57.8%	55.6%	49.1%	-
5-scale	*RandCrop*	54.1%	54.8%	53.1%	48.2%	39.0%
	*ActCrop*	59.4%	60.0%	60.0%	57.8%	49.3%

**Table 3 sensors-21-06774-t003:** MaxOverlap proportions of action boxes from ETRI-Activity3D-LivingLab videos compared to those from ETRI-Activity3D videos according to an action and a classifier.

Method	Action #	<0.6	<0.7	<0.8	<0.9	<1.0
Baseline	1	0.034	0.067	0.092	0.160	0.647
	8	-	0.059	0.119	0.110	0.712
	44	-	-	-	0.067	0.933
	48	-	-	0.042	0.025	0.932
Proposed	1	-	-	-	-	1.000
	8	-	-	0.017	0.085	0.898
	44	-	-	-	0.008	0.992
	48	-	-	-	-	1.000

**Table 4 sensors-21-06774-t004:** Action recognition results tested on the ETRI-Activity3D-LivingLab videos by the baseline and the proposed method trained with the ETRI-Activity3D or KIST SynADL videos. To reduce the influence due to the domain shift, ref. [46] is employed for UDA (Unsupervised Domain Adaptation). S and T are Source and Target domains, respectively.

Training Data	Method	Baseline	Proposed
(1) KIST SynADL	Supervised	32.5%	34.2%
(2) KIST SynADL (S) + ETRI-Activity3D (T)	UDA [46]	42.7%	43.9%
(3) ETRI-Activity3D	Supervised	50.4%	59.4%
(4) KIST SynADL + ETRI-Activity3D	Supervised	61.6%	67.2%

**Table 5 sensors-21-06774-t005:** MaxOverlap proportions of action boxes from ETRI-Activity3D-LivingLab video compared to those from KIST SynADL or ETRI-Activity3D according to an action and a classifier.

Training Dataset	Method	Action #	<0.6	<0.7	<0.8	<0.9	<1.0
KIST SynADL	Baseline	13	-	0.017	0.050	0.092	0.842
		18	-	-	0.008	0.083	0.908
		48	0.034	0.078	0.138	0.362	0.388
		52	-	-	0.025	0.076	0.898
	Proposed	13	-	0.017	0.050	0.075	0.858
		18	-	-	0.008	0.058	0.933
		48	-	-	0.008	0.034	0.958
		52	-	-	0.025	0.076	0.898
KIST SynADL + ETRI-Activity3D	Baseline	13	-	-	0.017	0.059	0.924
		18	-	-	-	0.008	0.992
		48	-	-	0.042	0.008	0.949
		52	-	-	-	0.025	0.975
	Proposed	13	-	-	-	0.008	0.992
		18	-	-	-	0.008	0.992
		48	-	-	-	0.025	0.975
		52	-	-	-	-	1.000

**Table 6 sensors-21-06774-t006:** Results of applying close-up of maximum activation to other existing methods. Only predictions for the videos of the size 171×128 are presented.

Method	1-Scale	2-Scale	3-Scale	4-Scale	5-Scale
Non-local	51.2%	54.7%	58.3%	60.1%	61.6%
SlowFast	50.3%	55.2%	59.1%	60.9%	62.3%

## Data Availability

Not applicable.

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
