# Peer review of "Action Recognition Using Close-Up of Maximum Activation and ETRI-Activity3D LivingLab Dataset"

_sensors, 2021, doi:10.3390/s21206774_

Round 1

Reviewer 1 Report

The paper presents an interesting subject. The following aspects must be updated and clearly presented:

  • remove from keywords list: (List three to ten pertinent keywords specific to the article; yet reasonably common within the subject discipline.)
  • abstract it is used ETRI-Activity3D7 LivingLab dataset and LivingLab dataset -> it must be clearly that both names represent the same dataset (also in Fig 1)
  • section Related Work must be renumbered in section 2 
  • section Related work must contains also results of the presented methods (eg. accuracy, inference time, etc)
  • figures must be places before their first appearance in the text (eg.Fig. 3)
  • explain more clearly what is the influence of position (x0,y0) that is randomly chosen for cropping 
  • how is chosen value for L 
  • in experimental results - it is not clearly what is the baseline model
  • caption of figure 7:"The maximum box is selected the an action box" - > must be reformulated 
  • for the experimental part:
    • all tested activities are improved in the same manner?
    • explain more clearly what should it happen in case of activities that have interactions with other objects and these objects are partially occluded or the illumination is low 
    • the ablation study with other models must be extended

Author Response

Thank you for your kind comments.

We revised our paper with the point-to-point responses.

Please see the attachment including the responses and the paper.

Reviewer 2 Report

The paper proposes a new ETRI-Activity3D-LivingLab dataset which can be used for generalization test 
and also proposes a close-up of maximum activation method in order to enhance action recognition process. 

The paper is well written, clear and focused taking into account proposed methods and results.

Some comments:

1- In Experimental results (5.1), the baseline is not defined and tlable 1 shoud appear after the corresponding text.

2- In table 2, in 4-scale results, the last numbers are not correct and must be chanced. If not should explain why.  
A comments should be included about the accuracy decreasing with increasing video dimentions.

3- In line 312 the phrase "The intra-dataset results in Table 2 are
approximately 35% lower in accuracy than the inter-dataset results in Table 1.
313", is not correct. Should be changed.

4- line 326 "comaprison" should be "comparison"

5 - In label of Figure 7, the phrase "The maximum box is selected
the an action box which contains all human boxes detected in a video by Yolo v3 [40].", must be re-written

Author Response

(The authors gave the same response as above.)

Round 2

Reviewer 1 Report

Since all my comments were addressed, I recommend to publish the paper.